# ^64^Cu-DOTHA_2_-PSMA, a Novel PSMA PET Radiotracer for Prostate Cancer with a Long Imaging Time Window

**DOI:** 10.3390/ph15080996

**Published:** 2022-08-13

**Authors:** Marie-Christine Milot, Ophélie Bélissant Benesty, Véronique Dumulon-Perreault, Samia Ait-Mohand, Patrick O. Richard, Étienne Rousseau, Brigitte Guérin

**Affiliations:** 1Department of Nuclear Medicine and Radiobiology, Faculty of Medicine and Health Sciences, Université de Sherbrooke, Sherbrooke, QC J1H 5N4, Canada; 2Sherbrooke Molecular Imaging Center (CIMS), Centre de Recherche du Centre Hospitalier Universitaire de Sherbrooke (CRCHUS), 3001, 12e Avenue Nord, Sherbrooke, QC J1H 5N4, Canada; 3Department of Surgery, Division of Urology, Faculty of Medicine and Health Sciences, University de Sherbrooke, Sherbrooke, QC J1H 5N4, Canada

**Keywords:** copper-64, DOTHA_2_ chelator, PSMA, prostate cancer, PET imaging, theranostic approaches

## Abstract

Prostate cancer imaging and late-stage management can be improved with prostate-specific membrane antigen (PSMA)-targeting radiotracers. We developed a PSMA positron emission tomography (PET) radiotracer, DOTHA_2_-PSMA radiolabeled with ^64^Cu (T_1/2_: 12.7 h), to leverage its large imaging time window. This preclinical study aimed to evaluate the biological and imaging properties of ^64^Cu-DOTHA_2_-PSMA. Its stability was assessed in plasma ex vivo and in mice. Cellular behavior was studied for up to 48 h in LNCaP cells. Biodistribution studies were performed in balb/c mice for up to 48 h. Dynamic (1 h) and static (4 h and 24 h) PET imaging was completed in LNCaP tumor-bearing mice. ^64^Cu-DOTHA_2_-PSMA was stable ex vivo in plasma and reached cellular internalization up to 34.1 ± 4.9% injected activity (IA)/10^6^ cells at 48 h post-injection (p.i.). Biodistribution results showed significantly lower uptake in kidneys than ^68^Ga-PSMA-617, our reference PET tracer (*p* < 0.001), but higher liver uptake at 2 h p.i. (*p* < 0.001). PET images showed ^64^Cu-DOTHA_2_-PSMA’s highest tumoral uptake at 4 h p.i., with a significant difference between blocked and non-blocked groups from the time of injection to 24 h p.i. The high stability and tumor uptake with a long tumor imaging time window of ^64^Cu-DOTHA_2_-PSMA potentially contribute to the prostate cancer theranostic approach and its local recurrence detection.

## 1. Introduction

Novel clinical tools are needed to improve prostate cancer detection [1,2]. Specifically, more sensitive and specific imaging agents are needed to improve staging for patients with localized, recurrent, or metastatic disease [2,3,4].

Positron emission tomography (PET) with tracers targeting the prostate specific membrane antigen (PSMA) has shown superiority or high potential in all states of the disease, including notably in metastatic identification at staging and in early biochemical recurrence [1,2]. Urea-based small-molecule radiotracers for PSMA PET showed high sensitivity and advantages over previously suggested PSMA antibodies, such as better tumor penetrability and faster blood clearance [1,2,5,6]. Many small PSMA molecules radiolabeled to various radionuclides have been developed and studied preclinically and clinically [1,2,5,6,7,8]. Gallium-68 (^68^Ga) PSMA-11 is the most widely used PSMA radiotracer for imaging [1]. A theranostic variant, ^68^Ga-PSMA-617, was developed using a DOTA chelator also able to bind therapeutic radiometals such as lutetium-177 and actinium-225 [9]. Radiotracers labeled to fluor-18 (^18^F), such as ^18^F-DCFPyl and ^18^F-PSMA-1007, are also increasingly used to benefit from the longer ^18^F half-life and shorter positron range as compared with ^68^Ga [1]. A recent network meta-analysis by Alberts et al. in recurrent prostate cancer concluded that there are only small differences between the three most commonly used PSMA tracers in clinical practice, ^68^Ga-PSMA-11, ^18^F-DCFPyl, and ^18^F-PSMA-1007 [6]. In terms of detection rate, they all exhibited objective superiority compared with other prostate cancer non-PSMA-targeting PET tracers, carbon-11 choline and ^18^F-fluciclovine [6].

Although promising and efficient PET PSMA tracers have already been developed and studied, it is still possible to improve the clinical arsenal with tracers offering a longer imaging time window. The use of the positron emitter copper-64 (^64^Cu) for the radiolabeling of PSMA tracers could offer an interesting addition (β^+^: 17.4%, average energy: 0.28 MeV, maximal energy: 0.65 MeV). For example, the 12.7 h half-life of ^64^Cu would allow plenty of time for manufacturing, quality control of radiotracers, easy exportation, and imaging [10,11,12], while being short enough for PET imaging with acceptable patient dosimetry [10,11]. Moreover, spatial resolution is theoretically better with ^64^Cu than with ^68^Ga radiotracers because of the shorter positron range and less frequent emission of gamma rays [10]. ^64^Cu could also serve in theranostic pairing with copper-67 for β-endoradiotherapy [10].

PSMA-targeting radiotracers using ^64^Cu have been proposed [13,14,15,16,17,18,19,20]. ^64^Cu-PSMA-617 showed a detection ability for local and distant prostate cancer lesions at initial diagnosis and biochemical recurrence in observational studies [21,22,23], but the release of free ^64^Cu from its DOTA chelator led to high liver uptake in preclinical studies, and metabolites were observed [13,14,15,16]. Other, more stable ^64^Cu-PSMA tracers have been suggested. Of these, ^64^Cu-CB-TE2A-PSMA from Banerjee et al. and ^64^Cu-CC34 from Gourni et al. showed improved stability and tumoral contrast, with a high kidney uptake [16,17]. Zia et al. and Dos Santos et al. subsequently suggested two other promising compounds: ^64^Cu-SAR-bisPSMA, currently studied in clinical settings (COBRA NCT05249127), and ^64^Cu-CA003, respectively [15,18].

Molar activities of ^64^Cu-PSMA tracers range from 2.9 to 50 MBq/nmol, and the reported radiolabeling method requires either a long labeling period (up to 60 min), high temperatures (up to 95 °C), or both [13,14,15,16,17,18,19,20]. Furthermore, even if the ^64^Cu-PSMA radiotracers suggested later showed promising tumoral contrast, distribution and clearance profiles are generally similar for all compounds, with important kidney uptakes [13,14,15,16,17,18,19,20].

Here, we developed a novel ^64^Cu-PSMA tracer and completed its preclinical characterization. We conjugated the hydroxamate bifunctional chelator DOTHA_2_ developed by our group [24] to the Glu-urea-Lys binding motif through the 2-naphthyl-L-Ala-AMCH linker and ^64^Cu to yield ^64^Cu-DOTHA_2_-PSMA (Figure 1). DOTHA_2_ is known for its rapid ^64^Cu complexation kinetic under mild conditions and complexation stability [24]. It demonstrated its potential in preclinical PET with a bombesin-based radiotracer showing favorable pharmacokinetic and improved tumor-to-normal tissue ratios over time [25].

We hypothesized that ^64^Cu-DOTHA_2_-PSMA could allow specific and high tumor contrast imaging of prostate cancer with a long imaging time window. Here, we evaluated the stability of ^64^Cu-DOTHA_2_-PSMA and assessed its in vitro behavior. We investigated its in vivo distribution and determined its tumor uptake, specificity, and excretion profile by mice PET imaging. Comparative studies were performed with ^68^Ga-PSMA-617, a reference PSMA tracer under clinical trial in our research center [9], and comparison with other ^64^Cu-PSMA tracers from the literature was discussed [13,14,15,16,17,18,19,20]. Briefly, ^64^Cu-DOTHA_2_-PSMA showed good stability and allowed PET imaging with high tumoral uptake up to 24 h post-injection (p.i.). Kidney uptake was lower than other PSMA tracers and liver uptake was higher (e.g., similarly to ^18^F-PSMA-1007). ^64^Cu-DOTHA_2_-PSMA has the potential for clinic translation for prostate bed imaging.

## 2. Results

### 2.1. Synthesis, Radiolabeling, and Characterization

#### 2.1.1. Synthesis of DOTHA_2_(OtBu)_3_

DOTHA_2_(OtBu)_3_ was synthesized in solid phase according to the protocol developed by our group [24] using a *N*-methyl-*O*-*tert*-butylhydroxylamine pendant arm [26] (Figure 1). Selective cleavage of the bifunctional chelate was performed using trifluoroethanol (TFE) in DCM, yielding a ready-for-coupling derivative DOTHA_2_(O*t*Bu)_3_ in good yield (43%) after purification (Figure 1, Appendix A).

#### 2.1.2. Synthesis, Radiolabeling, and Characterization of ^64^Cu-DOTHA_2_-PSMA

Standard amide coupling using *N*-(3-dimethylaminopropyl)-*N*’-ethylcarbodiimide hydrochloride (EDC)/*N*-hydroxysuccinimide (NHS) and *N,N*-diisopropylethylamine (DIPEA) was performed in *N*,*N*-dimethylformamide (DMF) to conjugate DOTHA_2_ (*O*tBu)_3_ to the PSMA ligand, followed by simple deprotection of the tri-*tert*-butyl esters to obtain the desired DOTHA_2_-PSMA with a 32% overall yield and purity >98% (Figure 1, Appendix A). The precursor was successfully radiolabeled with ^64^Cu(OAc)_2_ with a radiochemical yield higher than 99% (determined by UPLC and radio-TLC, (Appendix A)) in 5–10 min at room temperature without further purification and with excellent effective molar activity (116 ± 30 MBq/nmoL, *n* = 10) (Figure 1). The log *D* value of ^64^Cu-DOTHA_2_-PSMA was determined to be -0.96 ± 0.61 from the octanol phosphate-buffered saline (PBS) partition coefficient measurements.

#### 2.1.3. Preparation of ^68^Ga-PSMA-617

This tracer [9] is routinely prepared in our center with a radiochemical yield higher than 95%, as determined by UPLC (Appendix A).

### 2.2. Stability Studies

#### 2.2.1. Plasma Stability (Ex Vivo)

The stability of ^64^Cu-DOTHA_2_-PSMA was measured in mouse serum for 1 h, 4 h, and 24 h of incubation at 37 °C. No major demetallation or degradation of ^64^Cu-DOTHA_2_-PSMA was detected for up to 24 h, as monitored by radio-UPLC and radio-TLC (Table 1, Appendix A). In plasma supernatant, the signal from free ^64^Cu represented 0%, 1.98% and 1.28% of the total signal after 1 h, 4 h, and 24 h of incubation, respectively. The proportions of the ^64^Cu signal bound to plasma proteins (precipitate) were 24%, 34%, and 36% after 1 h, 4 h, and 24 h of incubation, respectively. From this signal, a portion was attributed to free ^64^Cu, which represented 0.79%, 0.64%, and 0.65% of the total signal at 1 h, 4 h, and 24 h of incubation, respectively. This was determined by treating the plasma protein precipitate with 10% trifluoroacetic acid (TFA) and analyzing the extracted signal. Overall (supernatant and precipitate), 0.79%, 2.62%, and 1.93% of the signal was attributed to free ^64^Cu following 1 h, 4 h, and 24 h of incubation, respectively.

When adding an excess of ^nat^Cu-DOTHA_2_-PSMA (Table 1), which would be used as a blocking agent for imaging study, the signal bound to plasma proteins slightly changed by 1% at 4 h and 6% at 24 h. In this context, the proportion of free ^64^Cu was also higher. This was 0.82%, 4.19%, and 7.75% after 1 h, 4 h, and 24 h of incubation, respectively.

We compared these results with the incubation of free ^64^Cu(OAc)_2_. In this context, the proportion of the signal bound to plasma proteins was greater than 90%. Furthermore, when treating the portion bound to plasma proteins with 10% TFA, all the free ^64^Cu signal could be extracted.

#### 2.2.2. In Vivo Stability

To evaluate stability in vivo, blood, urine, and liver samples were obtained after the injection of ^64^Cu-DOTHA_2_-PSMA in balb/c mice. In vivo, 48% of ^64^Cu-DOTHA_2_-PSMA was bound to plasma proteins at 1 h p.i. A total of 4.58% of ^64^Cu dissociation was observed in combined supernatant and plasma proteins at 1 h p.i. (Table 1, Appendix A). The intact ^64^Cu tracer was found in urine at 1 h p.i. (Appendix A). A total of 25% of a ^64^Cu metabolite was observed in the liver after 2 h p.i. (Appendix A). The signal was too low in blood and urine to enable later evaluation.

### 2.3. Cellular Assays

#### 2.3.1. Competition Assays

^nat^Cu-DOTHA_2_-PSMA and PMPA-2 inhibitory concentration 50 (IC_50_) values determined from average and standard deviations in triplicate on the human prostate cancer cell line LNCaP in competition with ^64^Cu-DOTHA_2_-PSMA and ^68^Ga-PSMA-617 are shown in Table 2. ^nat^Cu-DOTHA_2_-PSMA showed a good affinity of 11.3 ± 14.3 nM on LNCaP cells (entry 1). The IC_50_ of PMPA-2 in competition with ^64^Cu-DOTHA_2_-PSMA was not significantly different from that determined in competition with ^68^Ga-PSMA-617 (*p* = 0.530) (entries 2 and 3).

#### 2.3.2. Uptake, Internalization, and Efflux Assays

For the in vitro assays, the ^64^Cu-DOTHA_2_-PSMA results are shown in Figure 1 and in Appendix A. The ^64^Cu-DOTHA_2_-PSMA uptake increased from 22.0 ± 8.8 to 34.5 ± 13.6% injected activity (IA)/10^6^ cells from 30 min to 48 h p.i. (*n* = 9). It was significantly higher than the ^68^Ga-PSMA-617 uptake (*p* < 0.001). Over the first 2 h, ^64^Cu-DOTHA_2_-PSMA internalization did not significantly differ from ^68^Ga-PSMA-617 internalization (*p* = 0.42). At 2 h, ^64^Cu-DOTHA_2_-PSMA internalization reached 12.2 ± 6.7% IA/10^6^ cells, then increased to 34.1 ± 4.9% IA/10^6^ cells at 48 h p.i. ^64^Cu-DOTHA_2_-PSMA efflux was not significantly different from that of ^68^Ga-PSMA-617 (*p* = 0.95). The retention stabilized at approximately 42% from 8 h to 48 h p.i. (34.3–51.9%). Cell assays with too low or irregular confluence were rejected.

### 2.4. Animal Studies

#### 2.4.1. Balb/c Mice Biodistribution

^64^Cu-DOTHA_2_-PSMA biodistribution in balb/c mice, presented in Figure 2 and Appendix A, showed at most 15.8 ± 7.0% IA/g of uptake in the kidneys. The kidney uptake was significantly lower than with ^68^Ga-PSMA-617, with a difference of 77.9% and 23.3% at 1 h and 2 h p.i., respectively (*p* < 0.001) as shown in Figure 2b. In the liver, the uptake of ^64^Cu-DOTHA_2_-PSMA peaked at 30.2 ± 4.2% IA/g at 4 h p.i. At 2 h p.i., this was significantly higher than for ^68^Ga-PSMA-617 (29.9 ± 5.0% IA/g vs. 0.67 ± 0.16% IA/g, *p* < 0.001). At 2 h p.i., the ^64^Cu-DOTHA_2_-PSMA uptake was significantly higher in all organs except blood, adrenals, and fat, but with absolute differences of lower than 9% IA/g.

#### 2.4.2. PET Imaging

Representative ^64^Cu-DOTHA_2_-PSMA PET images are shown in Figure 3, and time–activity curves and histograms are shown in Figure 4 (Appendix A). On PET images, tumor uptakes at 1 h, 4 h, and 24 h p.i. were 14.0 ± 5.0% IA/cc, 23.8 ± 11.5% IA/cc, and 18.5 ± 6.6% IA/cc, respectively. With the co-injection of blocking agent, tumor uptake dropped to 4.78 ± 0.96% IA/cc, 7.46 ± 1.07% IA/cc p.i. and 8.62 ± 0.74% IA/cc at 1 h, 4 h, and 24 h, respectively, confirming the specificity of ^64^Cu-DOTHA_2_-PSMA at the tumor (with linear model of uptake vs. co-injection: *p* < 0.001 (Appendix A), 4 h: *p* = 0.004 and 24 h: *p* = 0.004).

The kidney uptake peaked at 5 and 10 min p.i. in the cortex and calyxes (Figure 4c,d), respectively. The signal in the kidney cortex was significantly lower with co-injection of the blocking agent for time frames from 10 min to 25 min and 40 to 45 min (includes signal from 5 to 25 min p.i. and 35 to 45 min p.i. (all *p*-values < 0.05)). In the calyxes, the differences were significant at 10, 20, and 25 min p.i. On static images at 4 h and 24 h p.i., kidney uptake was too low for ROIs to be drawn.

Liver uptake increased steadily and peaked at 4 h p.i. (26.8 ± 5.3% IA/cc). It was only significantly different with or without co-injection of blocking agent at 24 h p.i. (*p* = 0.04) (Figure 4e,f). Muscle uptake was stable from 5 min to 24 h p.i., between 1% and 2% IA/cc, without a significant difference between blocked and non-blocked injection.

The ratios of tumor signal to muscle, liver, and kidney signals and intervals are presented in Appendix A. The tumor-to-muscle ratios were 12.8, 21.3, and 11.9 at 1 h, 4 h, and 24 h p.i., respectively. The signal of the tumor to the cortex was 1.45 at 1 h p.i. It was not determined at 4 h and 24 h p.i. because the kidney signal was too low to allow the cortex ROI to be precisely drawn. The tumor signal was greater than the kidney signal. The tumor-to-liver ratios were lower, with values of 0.62, 0.89, and 0.87 at 1 h, 4 h, and 24 h p.i.

#### 2.4.3. Tumor-Bearing Mice Biodistribution

Biodistribution in tumor-bearing PET mice after imaging at 24 h p.i. corroborated PET data by displaying a similar distribution (Figure 4i, Appendix A). The signal was significantly lower with the co-injection of the blocking agent in some healthy organs (i.e., kidneys, testes, fat, lungs, brain, and salivary glands). The lower LNCaP tumor signal at 24 h p.i. with the blocking co-injection was not significantly different from the signal without (*p* = 0.218).

## 3. Discussion

### 3.1. Synthesis, Radiolabeling, and Characterization

The present study aimed determine the potential of ^64^Cu-DOTHA_2_-PSMA as a new prostate cancer PET imaging agent. High yield and excellent molar activities (116 ± 30 MBq/nmol) were obtained for the ^64^Cu-DOTHA_2_-PSMA conjugate, with rapid radiolabeling at room temperature under mild conditions. These three characteristics distinguish it from previously suggested ^64^Cu-PSMA tracers with lower molar activities (2.9 to 50 MBq/nmoL) and, for some of them, lower yield or more difficult radiolabeling [13,14,15,16,17,18,19,20].

### 3.2. Stability

Stability was subsequently evaluated. ^64^Cu-DOTHA_2_-PSMA proved to be fairly stable in vitro, ex vivo, and in vivo, as expected by the strong chelation property of DOTHA_2_ for ^64^Cu [24]. After 24 h of incubation in plasma, less than 2% of the ^64^Cu was free. After injection in mice, no metabolite was observed in urine, but 5% of free ^64^Cu was found in blood 1 h p.i. In the liver, no free ^64^Cu was observed at 2 h p.i., but the presence of a new radioactive peak was seen by radio-TLC. This probably corresponded to a ^64^Cu metabolite, whose identity was not determined. In comparison, ^64^Cu-PSMA-617 showed significant radiolabeled metabolites in blood and urine early after injection (i.e., 71% of the signal corresponded to the intact product in serum after 5 min [14] or 20%, 28%, and 5.2% of the signal corresponded to the intact product at 2 h p.i. in blood, liver, and urine, respectively [13]).

### 3.3. Cellular Assays

Cellular competition assays showed that the IC_50_ of ^nat^Cu-DOTHA_2_-PSMA was in the low nanomolar range, displaying a slightly higher affinity than the inhibitor PMPA-2 to PSMA (Table 2). The IC_50_ of ^nat^Cu-DOTHA_2_-PSMA was comparable to the results obtained with PSMA-617 (e.g., K_i_ = 2.34 ± 2.94 nM), although variability in results was observed [9]. Notably, the presence of bovine serum albumin (BSA, 20 g/L) in the RPMI medium used for binding assay contributed to increasing our IC_50_ values (i.e., 21.4 nM with 10% BSA in RPMI vs. 9.08 nM with RPMI only).

Cellular uptake values of ^64^Cu-DOTHA_2_-PSMA were significantly higher than those of ^68^Ga-PSMA-617. Internalization values were not significantly different when compared with ^68^Ga-PSMA-617’s internalization up to 2 h p.i. ^64^Cu-DOTHA_2_-PSMA’s internalization values after 2 h p.i. reached high levels and kept increasing up to values similar to the total uptake. This showed that the product enters the cell in important proportions after sufficient time of incubation (4 h and more). These results are promising for high tumoral signals in PET. For retention (efflux assay), the lower late values (8 h p.i. and later) as well as the important standard deviations on uptake and retention results could be partially due to technical issues (i.e., poor attachment of cells to the plates). Indeed, we noticed cell detachment occurring, and it could vary between the wells of a given experiment. This could also have affected the competition assays. Internalization assays were less affected, likely because the acid wash favored the release of most poorly attached cells and left uniform wells. The reports of uptake, internalization, and efflux at several time points were of interest to highlight the radiotracer cellular behavior. However, this method was perhaps more likely to be affected by detachment given the more frequent manipulation of plates. Notably, cellular viability was confirmed, and cellular passage and confluences were within acceptable ranges. We suggest the use of well filters to decrease cell loss for future experiments.

### 3.4. Animal Assays

#### 3.4.1. Biodistribution

In balb/c biodistribution, ^64^Cu-DOTHA_2_-PSMA demonstrated low renal uptake, in contrast to the experimental results with ^68^Ga-PSMA-617. It also differed from results from the literature for the majority of ^64^Cu- and ^68^Ga-based PSMA radiotracers [9,13,14,15,16,17,18,19,20]. A notable exception is the more important hepatobiliary clearance of the clinically used agent, ^18^F-PSMA-1007 [27]. The biodistribution of ^64^Cu-DOTHA_2_-PSMA was comparable with that of ^64^Cu-PSMA-617, but with higher uptake values at 24 h, notably in the liver and gastrointestinal tract (i.e., ^64^Cu-PSMA-617’s liver uptake of 9–13 %IA/g decreasing over 24 h) [13,14]. The high liver uptake of ^64^Cu-DOTHA_2_-PSMA could be explained by its greater lipophilicity (log *D* = −0.96 ± 0.61 versus −1.98 for ^68^Ga-PSMA-617 and −1.93 ± 0.13 for ^64^Cu-PSMA-617 [14,28]) and its binding to plasma proteins that should favor hepatic radiotracer elimination [29]. We cannot exclude the release of free ^64^Cu from ^64^Cu-DOTHA_2_-PSMA, which could also partially explain the high liver uptake observed [13,14,15,16]. Even if 5% of free ^64^Cu was observed in blood at 1 h p.i., free ^64^Cu was not seen in liver at 2 h p.i. It was not possible to confirm the presence of ^64^Cu in the blood, liver, and urine at later time points in mice; the radioactive counts were too low to enable evaluation.

#### 3.4.2. PET Imaging

PET imaging showed high tumoral uptake peaking at 4 h p.i., which seemed to be the optimal imaging time-point for ^64^Cu-DOTHA_2_-PSMA, considering the maximal tumor signal and tumor-to-non-specific tissue ratios. ^64^Cu-DOTHA_2_-PSMA tumoral uptake and tumor-to-muscle ratios remained high until 24 h p.i., offering high tumor contrast with a long imaging time window. From 1 h to 24 h p.i., tumor uptake and tumor-to-muscle ratios of ^64^Cu-DOTHA_2_-PSMA were similar to those of ^64^Cu-CA003, ^64^Cu-SarbisPSMA, and ^68^Ga-PSMA-617 (only 1 h p.i. available). The tumor-to-muscle ratio of ^64^Cu-DOTHA_2_-PSMA was lower than for ^64^Cu-CC34 and ^64^Cu-CB-TE2A-PSMA from 1 h to 24 h p.i. (studied in the transfected tumor overexpressing PSMA [30]) [9,15,16,17,18]. The co-injection of ^nat^Cu-DOTHA_2_-PSMA significantly decreased ^64^Cu-DOTHA_2_-PSMA tumoral uptake from injection to 24 h p.i., suggesting the specificity of the signal. ^nat^Cu-DOTHA_2_-PSMA was selected for blocking study in order to yield a similar pharmacokinetic to the radiotracer studied, in opposition to more hydrophilic or rapidly cleared compounds (e.g., PMPA-2 [31,32]). No significant decrease in the tumoral signal was observed in the NRG biodistribution results at 24 h with the blocking co-injection. This could be partially explained by the variation in the tumor weight: in biodistribution, the whole tumoral bulk is harvested, including contained bleeding or dead regions undistinguishable to the eye, while some regions of LNCaP tumors with a jelly consistency can be lost. These effects lead to an increase in tumor weight variability, and therefore, variability in %IA/g for the tumors.

Regarding the kidneys and the urinary clearance, time–activity curves showed that the urinary clearance of ^64^Cu-DOTHA_2_-PSMA measured by kidney calyxes and cortex signals peaked at 10 min or less. The tumor-to-kidney ratio favored tumors at 1 h p.i. and increased at later time points (increase in tumor signal and decrease in kidney signal), although the ratio could not be precisely calculated in later time points. We analyzed the cortex and calyxes separately on PET images to consider their different retention mechanisms: the calyx signal is highly unspecific because it mainly collects excreted urine; meanwhile, the cortex uptake is partially specific due to the presence of the PSMA enzyme in tubules in addition to unspecific urine collection [33]. The renal medulla can be present in part in both calyx and cortex ROIs. The uptake in the PSMA-positive kidney cortex was significantly inhibited at early time points (5–25 min, 35–45 min). Difficulties in observing blocking in the kidney cortex at 1 h p.i. could be the results of the low uptake of ^64^Cu-DOTHA_2_-PSMA. Other preclinical studies showed higher kidney uptake (i.e., 67% to 275% IA/g at 1 h p.i.) and observed important ability to block kidney uptake with their different blocking protocols [9,15,16,17]. ^64^Cu-DOTHA_2_-PSMA is the only agent with tumor-to-kidney ratios greater than 1 from 1 h p.i., with other compounds requiring 4 h to 48 h of clearance to reach ratios greater than 1 [9,13,14,15,16,17,18,19,20]. It is to be emphasized that for the current study and for all reported studies, the divergence between murine PSMA and human PSMA is a limitation in murine preclinical studies and could lead to different signal ratios of the tumor to kidneys and other PSMA-expressing organs in humans [33].

As for the liver and hepatobiliary circulation, time–activity curves showed that liver and bowel clearance represented an important excretion path for ^64^Cu-DOTHA_2_-PSMA, as anticipated from biodistribution studies. The tumor-to-liver ratios were low at all time points. Other ^64^Cu and ^68^Ga PSMA radiotracers have a lower hepatic signal with higher tumor-to-liver ratio from time of injection [9,15,16,17,18,19,20]. ^64^Cu-PSMA-617 also has a lower hepatic uptake than ^64^Cu-DOTHA_2_-PSMA, but its tumor-to-liver ratios are lower given its low tumoral uptake (based on the results from dos Santos et al. [15]). In the present study, there was a small reduction in signal in the liver for ^64^Cu-DOTHA_2_-PSMA at 4 h p.i. (non-significantly) and at 24 h p.i. (significantly) with the co-injection of a blocking agent. The blocking could be mediated by an effect other than PSMA expression because it was also observed in the muscles (non-significantly) and because PSMA is not expressed in the liver [33]. Ex vivo stability studies showed that the co-incubation of the blocking agent ^nat^Cu-DOTHA_2_-PSMA reduced the binding to plasma proteins from 36% to 30% at 24 h p.i. (Table 1). Based on these results, the interactions between plasma proteins and ^64^Cu-DOTHA_2_-PSMA appear to be reversible. The co-injection of ^nat^Cu-DOTHA_2_-PSMA could release the radioactive PSMA conjugate from plasma proteins in blood circulation and modify its excretion profile. The plasma protein/^64^Cu-DOTHA_2_-PSMA complex of high molecular weight could mainly be eliminated by the liver, but free ^64^Cu-DOTHA_2_-PSMA released by the blocking agent competition for plasma proteins could be eliminated rapidly by the kidneys, as seen in the time–activity curves, contributing to a reduced liver uptake. These findings are consistent with the results in the literature [34]. Drugs that bind to plasma proteins can potentially be displaced by drugs with similar or greater affinity for the same binding site. In summary, these effects of the blocking co-injection on plasma proteins’ binding and clearance could explain the significant variation in the liver signal at 24 h p.i. in PET imaging, as well as other significant modifications in the signal in 24 h biodistribution results in non-PSMA-expressing organs (e.g., fat).

### 3.5. Implications and Future Research

In a broader context, ^64^Cu-DOTHA_2_-PSMA showed potential for PET imaging and precision medicine due to its tumoral uptake and contrast. It also has logistical advantages, including in comparison with other ^64^Cu-PSMA tracers, such as its rapid radiolabeling at room temperature with high molar activity and stability. ^64^Cu-DOTHA_2_-PSMA’s rapid urinary clearance and low activity in the urinary system combined with long-lasting high tumor uptake could enable better visualization of the prostate bed after voiding. This could enable higher local sensitivity to detect tumoral lesions with a long imaging time window, e.g., for local biochemical relapse imaging.

Current, most clinically used PSMA radiotracers are ^68^Ga-PSMA-11, ^18^F-PSMA-1007, and ^18^F-DCFPyL, which show superiority to conventional imaging and other PET radiotracers, such as fluciclovine and cholines in recurrent prostate cancer [6]. ^68^Ga-PSMA-11 and ^18^F-DCFPyL are mainly cleared by the kidneys [27]. In contrast, ^18^F-PSMA-1007 has a more important hepatobiliary clearance, similarly to ^64^Cu-DOTHA_2_-PSMA, and is considered to have potential for prostate cancer bed imaging [27]. Furthermore, from a theranostic perspective, a longer time window than for ^68^Ga- and ^18^F-based current PSMA tracers thanks to the half-life of ^64^Cu could also enable PET-imaging-based dosimetry for radionuclide therapy selection and planning. ^64^Cu-DOTHA_2_-PSMA importantly binds plasma proteins (48% in vivo), which could have the advantage of extending its blood retention time and maximizing its uptake in tumors. ^64^Cu-DOTHA_2_-PSMA’s rapid urinary clearance pattern also decreases kidney and bladder wall irradiation. The important liver uptake and hepatobiliary excretion could be a challenge for the identification of liver metastases in advanced prostate cancer. On the upside, it could be advantageous to have PSMA radiotracers with different clearance profiles from previous PSMA radiotracers to adapt according to patient comorbidities (e.g., renal failure and liver failure) and previous irradiation (e.g., pelvic radiotherapy). Further research is needed to investigate uptake in PET imaging in humans, because hepatic metabolism and tumoral uptake can diverge from preclinical studies, and to find solutions for metastatic lesion diagnostics, e.g., by using parametric imaging in the liver.

## 4. Materials and Methods

### 4.1. Chemistry

#### 4.1.1. General

All chemicals and solvents were used as supplied by the vendors without further purification unless otherwise stated. 2-Chlorotrityl chloride resin was obtained from Chem-Impex International Inc. (Wood Dale, IL, USA). Fmoc-protected amino acids were obtained from either EMD NovaBiochem (Gibbstown, NJ, USA) or Chem-Impex International Inc. (Wood Dale, IL, USA). Cyclen was obtained from CheMatech (Dijon, France). Acetonitrile (CH_3_CN), dichloromethane, DMF, and methanol were obtained from Fisher Scientific (Ottawa, ON, CA). DMF was dried over 4 Å molecular sieves for at least one week before its use to remove trace amounts of amines, and it was filtered before its use. All instruments were calibrated and maintained with standard quality-control procedures. Mass spectra were recorded on an API 3000 LC/MS/MS (Applied Biosystems/MDS SCIEX, Concord, Ontario Canada), on a Waters/Alliance HT 2795 equipped with a Waters 2996 PDA, and a Waters Micromass ZQ detector API 2000 on ESI-Q-Tof (MAXIS). High-resolution mass spectrometry was carried out through electrospray ionization using a Triple TOF 5600 ABSciex mass spectrometer. The purification of the DOTHA_2_-PSMA conjugate was assessed on a Biotage HPFC SP4 Flash Purification System and a C18 column. Analytical HPLC was performed on an Agilent 1200 system (Agilent Technologies, Mississauga, ON, L5N 5M4, Canada) equipped with a Zorbax Eclipse XDB C18 reversed-phase column (4.6 × 250 mm, 5 μ) and an Agilent 1200 series diode array UV-Vis detector (Agilent Technologies) using the following method: flow = 1 mL/min; 0–23 min; 0 to 76.6% CH_3_CN-0.05% TFA in H_2_O-0.05% TFA, 23–24 min; 100% CH_3_CN, 24–30 min; 100 to 0% CH_3_CN in H_2_O. HPLC analysis of ^68^Ga-PSMA-617 was performed on an Agilent 1100 system equipped with a Phenomenex column (4.6 × 250 mm, 5 μ) and an Agilent 1100 series diode array UV–Vis detector (Agilent Technologies) using the following method: flow = 1.2 mL/min; 15–50% CH_3_CN-0.05% formic acid in H_2_O-0.05% formic acid. ^64^Cu was prepared on a TR-19 or a TR-24 cyclotron (ACSI) by the ^64^Ni(p,n)^64^Cu reaction using an enriched ^64^Ni target electroplated on a rhodium disc [35]. ^64^CuCl_2_ was recovered from the target following the procedure of McCarthy et al. and converted to ^64^Cu(OAc)_2_ by dissolving the ^64^CuCl_2_ in ammonium acetate (0.1 M; pH 5.5) [36].

#### 4.1.2. Synthesis of DOTHA_2_ (OtBu)_3_

2-Bromo-O-*tert*-butyl-N-methylhydroxylamine-acetamide [26] (2.5 g, 11 mmol, 15 equiv.) and triethylamine (1 mL, 7.2 mmol, 9 equiv.) were dissolved in DMF (40 mL) and added to the cyclen-functionalized resin (1g, 0.74 mmol) [24]. The reaction was left shaking overnight at room temperature. The solvent was removed by filtration and the resin was washed with iPrOH (3×), H_2_O (3×), DCM (3×), MeOH (3×), and finally with diethylether (3×). A Kaiser test was performed to confirm the absence of free amines; the resin and solution were colorless to light yellow (negative test). The chelator was selectively cleaved by adding 5 mL of DCM/TFE (2/1) *v*/*v* to the resin and kept under vigorous magnetic stirring for 2 h at room temperature. The solution was subsequently filtered, and the resin was washed with DCM and MeOH. The combined filtrates were collected and reduced in a vacuum to yield a yellow oil. The residue was precipitated in diethyl ether to yield DOTHA_2_(O*t*Bu)_3_. The crude product was purified on a Biotage SP4 system equipped with a C18 column. The product fractions were pooled and lyophilized to obtain DOTHA_2_(O*t*Bu)_3_ in good yield (43%) as a beige solid. HPLC: Rt = 16.84 min (purity 94%). ^1^H NMR (400 MHz, CDCl_3_): δ_H_ (ppm) 4.09 (8 H, large s), 3.19–3.46 (25 H, m), 1.35 (27 H, s). ^13^C NMR (100 MHz, CDCl_3_): δc (ppm) 161.53, 161.17, 160.81, 160.45, 81.90, 54.50, 50.42, 39.63, 27.65. ESI-MS: calcd for C_31_H_61_N_7_O_8_ [M]^+^, 659.45; found [M + H]+, 660.62.

#### 4.1.3. Synthesis of DOTHA_2_-PSMA

The peptidomimetic glutamate–urea–lysine binding motif and the linker moiety were synthesized by solid-phase peptide chemistry with an overall yield of 67%, as previously described by Benešová et al. [37] For its conjugation, DOTHA_2_ (*O*tBu)_3_ [24] (50 mg, 0.076 mmol) was first activated by 1.1 equiv. of NHS (10 mg, 0.087 mmol) and 1.1 equiv. of EDC (16 mg, 0.086 mmol) in 600 μL of anhydrous DMF at 0 °C for 30 min and then at room temperature overnight (Figure 1). A twofold excess of activated DOTHA_2_ was added to the PSMA ligand (20 mg, 0.31 mmol) in DMF, mixed with 6 equiv. of DIEA (32 μL, 0.183 mmol). After stirring for 3 h at room temperature (Figure 1), the reaction mixture was evaporated to dryness. Subsequent removal of *t*Bu-protecting groups was carried out by dissolving the crude product in neat TFA and stirring for 24 h. After precipitation in cold diethyl ether, the crude DOTHA_2_-PSMA was dissolved in water and purified by flash chromatography on a Biotage SP4 system equipped with a C18 column. The product fractions were pooled and lyophilized to obtain the desired DOTHA_2_-PSMA in good yield (32%). Purity of the peptide was verified by HPLC and accounted for 98%; the identity was confirmed by API 3000 LC/MS/MS. HPLC: *R*_t_ = 14.65 min; ESI-MS: calcd for C_52_H_80_N_12_O_16_^+^, 1128.58; found, 1129.72 [M].^+^, 565 [M/2].^+^ (Appendix A).

#### 4.1.4. Preparation of ^nat^Cu-DOTHA_2_-PSMA

The ^nat^Cu-DOTHA_2_-PSMA was prepared by treating DOTHA_2_-PSMA (10 mg, 8.8 mmol) with a slight excess (1.1 equiv.) of trace metal copper acetate Cu(OAc)_2_ (1.8 mg, 9.6 mmol) in 3 mL of 0.1 M ammonium acetate buffer, pH 5.5 for 20 min at room temperature. During this time, the color of the solution turned from white to blue-green. The compound was purified on C18 sept Pak and eluted with ethanol, and isolated as a pale blue powder (9.5 mg, 8 mmol) of the desired complex with a yield of 90% and a purity of 97%. HPLC: *R*_t_ = 14.23 min; ESI-MS: calcd for C_52_H_78_CuN_12_O_16_ [M]^+^, 1190.81; found, 1191.72 [M+1]^+^, 596 [M/2].^+^ (Appendix A).

#### 4.1.5. Radiolabeling of DOTHA_2_-PSMA with ^64^Cu(OAc)_2_

Radiolabeling was performed by incubating DOTHA_2_-PSMA (3–5 nmol) with 220–500 MBq of ^64^Cu(OAc)_2_ in a total volume of 400 μL of 0.1 M ammonium acetate buffer, pH 5.5 within 5–10 min at room temperature. The radiolabeling was followed by radio-TLC on C18 plates using sodium citrate 0.1 M at pH = 5.5 as eluent. ^64^Cu-DOTHA_2_-PSMA was used without purification.

#### 4.1.6. Lipophilicity (log D) Measurements

For log *D* measurements, 8 MBq of ^64^Cu-DOTHA_2_-PSMA was added to a pre-saturated mixture of phosphate-buffered saline pH 7.4 (500 µL) and octanol (500 µL). Samples were shaken for 30 min at room temperature, centrifuged at 7000 rpm for 5 min, and 100 µL of each phase was counted using a Packard Cobra gamma counter. Experiments were performed in triplicate and the value was presented as log D_mean_ ± SD, where D was calculated as D = (Average of CPM in octanol / CPM in PBS).

#### 4.1.7. Preparation of ^68^Ga-PSMA-617

PSMA-617 was synthesized as reported in the literature [9]. ^1^H NMR (400 MHz, MeOD): δ_H_ 7.78 (m, 3H), 7.68 (brs, 1H), 7.41 (m, 3H), 4.66 (dd, 1H, ^1^J_HH_ = 6.4, ^2^J_HH_ = 8.4), 4.31 (dd, 1H, ^1^J_HH_ = 5.2, ^2^J_HH_ = 8.4), 4.17 (dd, 1H, ^1^J_HH_ = 4.7, ^2^J_HH_ = 8.4), 3.81 (brs, 8H), 3.29 (m, 17H), 3.10–3.15 (m, 5H), 2.42 (m, 2H), 2.14 (m, 2H), 1.21–1.91 (m, 14H), 0.93 (m, 2H). ESI-MS Calcd for C_49_H_7_1N_9_O_16_ [M]^+^: 1042,139. Found [M + H]^+^: 1042,4. The radiolabeling was performed by incubating PSMA-617 (20 nmol) with 1–1.2 GBq of ^68^GaCl_3_ (adjusted to pH 3.8 with 1 M ammonium acetate buffer, pH 4.5) at 95 °C for 12 min. The radiolabeling was followed by radio-TLC (C18 sodium citrate pH = 5.5 as eluent) and by radio-HPLC. ^68^Ga-PSMA-617 was obtained with high radiochemical purity (> 98%, ITLC and radio-HPLC, Appendix A) and was used without purification.

All compounds were > 95% pure by HPLC.

### 4.2. Stability Studies

#### 4.2.1. Plasma Stability (Ex Vivo)

Plasma stability studies were carried out by incubating ^64^Cu-DOTHA_2_-PSMA (50 MBq in 250 μL PBS buffer) in 250 μL of mouse serum at 37 °C for up to 24 h. The plasma was then mixed twice with ethanol (1:1) to precipitate all the proteins. The samples were subjected to vortex mixing for 1 min and then centrifugation for 10 min at 7000 rpm. Plasma proteins were separated from the soluble component by ultracentrifugation and the radioactivity was assessed. The supernatant fraction was assayed by radio-TLC on C18 plates and by UPLC; free ^64^Cu(OAc)_2_ and ^64^Cu-DOTHA_2_-PSMA were used as standards. The radio-TLCs were eluted with 0.1 M sodium citrate buffer at pH 5.5 using an Instant Imager system (BioScan, DC, U.S.A.) for radiodetection. The precipitated proteins were again subjected to 10% TFA, vortexed for 2–3 min, and then centrifuged for 10 min at 7000 rpm. The separation and analysis for the supernatant and the proteins were performed in the same manner.

The protein binding ability was evaluated by incubating ^64^Cu-DOTHA_2_-PSMA (100 MBq) in phosphate buffer (250 μL, pH 7) at 37 °C and fresh mouse serum (250 μL) and for 1 h, 4 h, and 24 h. Plasma proteins were precipitated by the addition of ethanol (1:1, *v*/*v*) to the plasma samples, followed by centrifugation at 7000 rpm for 5 min at room temperature and separation of the supernatant. This process was repeated three times with the same volume of ethanol, and the supernatants were pooled. The radioactivity content of the supernatant and the precipitated protein were measured in a dose calibrator to obtain the protein binding percentage.

#### 4.2.2. In Vivo Stability

For metabolite analysis, three balb/c mice were injected with a bolus of 32 MBq of ^64^Cu-DOTHA_2_-PSMA in 0.3 mL saline by tail vein (real IA: 26.4–39.0 MBq). Blood was collected at 1 h p.i. (0.01–0.1 mL) in heparin-coated tubes and then centrifuged at 10,000 rpm for 10 min. Subsequently, the pellet and supernatant were separated, and the relative activity was determined. Radio-TLC was performed to assess the stability of the radiolabeled compound in the blood and in plasma. Two balb/c mice were injected with 45 MBq (real IA: 44.7–45.5 MBq). Urine was collected at 1 h p.i. and analyzed by UPLC. Two balb/c mice were injected with 59 MBq and then euthanized and dissected at 2 h p.i. to harvest the liver (real IA: 58.9–59.3 MBq). The livers were promptly washed and put on ice. They were homogenized with an ice-cold piston with 1 mL of water, extracted, centrifuged (14,000 rpm; 3 min), and mixed with 1 mL of acetonitrile before 4 further cycles of centrifugation and the collection of supernatants. All the pellets were mixed with 1 mL of 10% TFA and centrifuged, and the final supernatant was analyzed by Radio-TLC. The activity was measured in all the supernatants.

### 4.3. Cellular Assays

#### 4.3.1. Cellular Model

Human prostate adenocarcinoma LNCaP PSMA-expressing cells were purchased from American Type Culture Collection (ATCC; CRL-1740, RRID:CVCL_1379). They were used for in vitro assays and for xenografts. PSMA expression was confirmed by immunohistochemistry in formalin-fixed paraffin-embedded LNCaP xenograft sections, stained with anti-PSMA antibody [YPSMA-1] (Abcam—ab19071, dilution 1:1500). Cells with fewer than 40 passages were used. The cells were cultured using RPMI-1640 media enriched with 10% fetal bovine serum, 1% penicillin–streptomycin, 1% amphotericin, and 1% l-glutamine (Wisent). All the experiments were performed using phosphate-buffered saline and trypsin (Wisent).

#### 4.3.2. Cellular Competition Assays

Here, 12-well polylysine-coated Cellbind plates were seeded with 150 × 10^3^ LNCaP cells. At 70–100% confluence (3 days growth), the cell culture medium was changed to 800 µL reaction medium (RPMI, 2% HEPES, 1% penicillin/streptomycin, 20 g/L bovine serum albumin). A total of 100 µL of ^64^Cu-DOTHA_2_-PSMA at 7.5 nM (0.068–0.091 MBq/well) was added at the same time as 100 μL of inhibitors at a concentration from 10^−3^ to 10^−13^ M. Inhibitors were PMPA-2 and ^nat^Cu-DOTHA_2_-PSMA. Cells were then incubated for 1 h, after which the medium was removed, cells were washed once with 1 mL of cold PBS, and then 1 mL of trypsin was added to collect cells for radioactivity measurement by gamma-counter. The assays were performed in triplicate, each time with independent ^64^Cu-DOTHA_2_-PSMA production and cell seedings. IC_50_ was calculated using GraphPad Prism 9 one site fit logIC_50_.

#### 4.3.3. Cellular Uptake, Internalization and Efflux Assays

Here, 12-well polylysine-coated Cellbind plates were seeded with 150 × 10^3^ LNCaP cells with 1.75 mL of cell media. At 70–100% confluence (3 days of growth), the cell medium was changed and 37 kBq of ^64^Cu-DOTHA_2_-PSMA in 1 mL of cell medium was added to each well of group 1 and 160 kBq to each of group 2.

Uptake and internalization assay cells were incubated for 15, 30, 45, 60 min, 2 h, 3 h, and 4 h in group 1 or 4 h, 8 h, 12 h, 24 h, 36 h, and 48 h in group 2 (3 wells per time point in all groups). Repetition of the 4 h time point was used to ensure reproducibility. After incubation, the medium was removed, and cells were washed with 1 mL of cold PBS. For the uptake assays, cells were directly treated with 1 mL of trypsin to be collected and their activity was assayed (HIDEX AMG Gamma Counter 425-601). For internalization assays, cells were incubated with 1 mL of acidic glycine (50 mM, pH = 2,8) for 5 min to remove membrane-bound ^64^Cu-DOTHA_2_-PSMA, washed with 3 mL of cold PBS, and collected using 1 mL of trypsin to be measured in the gamma-counter. Uptake and internalization results were expressed as the percentage of activity injected in the well (IA) in functions of time, normalized to 10^6^ cells.

For efflux assays, after adding ^64^Cu-DOTHA_2_-PSMA, cells were incubated for 60 min for the time points 15 min to 4 h (group 1) and incubated for 4 h for the time points 4 h to 48 h (group 2). After incubation, the medium was removed, cells were washed with 1 mL of cold PBS, 1 mL of cell media was added and, similarly to uptake and internalization, cells were again incubated in nonradioactive media for 0, 15, 30, 45, 60 min, 2 h, 4 h (group 1) and 4 h, 8 h, 24 h, 32 h or 48 h (group 2) (3 wells per time point). After respective incubation times per well, the medium was removed, and cells were washed with 1 mL of cold PBS, then removed from the well with 1 mL of trypsin to measure the residual radioactivity by gamma counter. The results of residual activity were expressed as percentage of the 1 h (Group 1) or 4 h (Group 2) uptake activity (0 min of efflux time-point) as a function of time for 10^6^ cells.

All the assays were performed at least 3 times, each with different ^64^Cu-DOTHA_2_-PSMA production and cell passage. Cells were counted in 3 wells for each protocol (uptake, internalization, and efflux) on day 1 and day 3 of each experiment and an average for cells numbers at day 2. In vitro experiments were compared with ^68^Ga-PSMA-617 results obtained following the same protocol for early time points (2 h p.i. and earlier), considering that ^68^Ga’s shorter half-life than ^64^Cu’s half-life did not enable later time points.

### 4.4. Animal Studies

#### 4.4.1. Animal Models

All the experiments were conducted according to a protocol approved by the Animal Ethics Committee of the Université de Sherbrooke and were in accordance with the guidelines of the Canadian Council on Animal Care. All animal manipulations that could cause pain, stress, or that required immobilization were performed under isoflurane anesthesia (2% in 2 L/min oxygen) with a heating pad or bed. When protocols required animal death (e.g., biodistribution) or ethical limits (e.g., tumor size) were reached, euthanasia was performed by CO_2_ inhalation under isoflurane anesthesia. Biodistributions and stability experiments did not require tumor-bearing animals and were conducted on male balb/c mice (4–8 weeks old and 20–26 g at time of experiment). For PET imaging, NRG mice were used (The Jack Laboratory, 7–10 weeks old, and 19–26 g at time of experiment) [38,39,40]. A total of 200 µL of a 1:1 Matrigel/PBS mixture containing 107 LNCaP cells was injected subcutaneously on each shoulder of NRG mice. Experiments were initiated 2 to 4 weeks after inoculation when tumors reached a diameter between 5 and 10 mm.

#### 4.4.2. Balb/c Mice Biodistribution

^64^Cu-DOTHA_2_-PSMA biodistribution was assessed in balb/c mice (*n* = 30) by harvesting blood and organs at 1 h, 2 h, 4 h, 24 h, and 48 h p.i. to measure tissue radioactivity and weight. Results were reported as the %IA/g of tissue and compared with the radiotracer used for clinical imaging in our facility, ^68^Ga-PSMA-617.

Balb/c mice (*n* = 30) were injected with a bolus of ^64^Cu-DOTHA_2_-PSMA by the tail vein (in 0.3 mL of saline: 3 MBq for 1 h, 2 h, and 4 h p.i., 7.5 MBq for 24 h p.i. and 21 MBq for 48 h p.i.). Injected activity had to be increased for later time points to ensure reliable detection and reliability in organs with a low uptake, e.g., the brain. At all timepoints, blood was collected using a heparin-coated syringe by femoral vein puncture and mice were then euthanized and dissected. Harvested tissues were rinsed, blotted dry, and then weighted and counted in gamma counter. The results are reported as the %IA per gram of tissue. The biodistribution results were compared with the ^68^Ga-PSMA-617 results obtained by our team at 1 h and 2 h p.i.

#### 4.4.3. PET Imaging

Dynamic (0–1 h p.i.) and static (4 h, 20 min long and 24 h p.i., 1h long) PET assays were performed in combination with 5 min CT in NRG mice injected with 8 MBq of ^64^Cu-DOTHA_2_-PSMA in 0.3 mL of saline through the tail vein (LabPET 8/-Triumph; Gamma Medica, Northridge, CA, USA). Blocking was achieved by the co-injection of 20 nmol of ^nat^Cu-DOTHA_2_-PSMA (*n* = 3 per group). We selected this blocking agent because its pharmacokinetics are similar to ^64^Cu-DOTHA_2_-PSMA. PET images were reconstructed using the three-dimensional maximum likelihood estimation method (MLEM-3D) algorithm with 20 iterations and a 60 mm field-of-view (Gamma Medica). The analyses were conducted using AMIDE software 1.0.4. Quantitative image data were reported as %IA per volume (centimeter cube, cc).

#### 4.4.4. Tumor-Bearing Mice Biodistribution

PET imaging of NRG mice and supplemental tumor-bearing mice were biodistributed 24 h p.i. as described for balb/c mice.

### 4.5. Statistics

The results were corrected for physical decay to the time of injection and are reported as the mean ± standard deviation, unless otherwise specified. Statistical significance was assessed by two-tailed Student’s t-tests (Holm– Šídák correction for repetitive measures, Welch test when applicable, adjusted *p* < 0.05 considered statistically significant), or by a linear model (code available as Appendix A) (programs: Excel, GraphPad Prism 8, R x64 4.1.1 and Python 3.8 using Spyder 4).

## Data Availability

The data presented in this study are available within this article and associated Appendix A.

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
