# Peer review of "64Cu-DOTHA2-PSMA, a Novel PSMA PET Radiotracer for Prostate Cancer with a Long Imaging Time Window"

_pharmaceuticals, 2022, doi:10.3390/ph15080996_

Round 1

Reviewer 1 Report

The manuscript titled “64Cu-DOTHA2-PSMA, a novel PSMA PET radiotracer for prostate cancer with a long imaging time window” by Marie-Christine Milot et al. reports the development of PSMA, PET radiotracer, DOTHA2-PSMA radiolabeled with 64Cu to influence its large imaging time window. The developed tracer is adequately characterized and studied for cellular behaviors and in vivo bio-distribution. The work is promising, the study is very systematic with sufficient supplementary data. A point could be considered in order to make the manuscript more reasonable and informative.

 1. Is the developed tracer enters the cell or are more concentrated on the periphery. May be data from Immunohistochemistry on LNCaP tumor could be useful.

Author Response

We thank the Reviewer 1 for his(her) overall positive assessment of our work and for constructive criticism. Please find our point-by-point responses to the reviewer’ comments below.

Reviewer 1. The manuscript titled “64Cu-DOTHA2-PSMA, a novel PSMA PET radiotracer for prostate cancer with a long imaging time window” by Marie-Christine Milot et al. reports the development of PSMA, PET radiotracer, DOTHA2-PSMA radiolabeled with 64Cu to influence its large imaging time window. The developed tracer is adequately characterized and studied for cellular behaviors and in vivo bio-distribution. The work is promising, the study is very systematic with sufficient supplementary data. A point could be considered in order to make the manuscript more reasonable and informative.

  1. Is the developed tracer enters the cell or are more concentrated on the periphery. May be data from Immunohistochemistry on LNCaP tumor could be useful.

Results from our cellular studies suggest that the majority of 64Cu-DOTHA2-PSMA was bound to the plasma membrane over the first 2h and was gradually internalized. After 4h of incubation, the cellular internalization of our radiotracer reached approximately half the total uptake and standard deviations were overlapping. The internalization kept increasing over 48h of incubation to reach the total uptake value. This was clarified in the discussion of the article (pp 8-9 of 19) .

Also, we have performed immunohistochemistry on LNCaP tumor to confirm PSMA expression and the result is presented in section 4.3.1. We cannot use the technique to determine if the developed tracer enters the cell or is more concentrated on the periphery. However, autoradiography with 64Cu-DOTHA2-PSMA may be more appropriate to determine the distribution of our radiotracer within the tumor. Unfortunately, the poor resolution of this technique will not allow visualization of intracellular organelles and the cytoskeleton.  

Reviewer 2 Report

The authors address a critical problem pertaining to prostate cancer imaging and have therefore designed and evaluated a novel PSMA targeted radiotracer. The authors have a good grasp on the methods performed and the manuscript is very well organized, has ideal studies performed and the results are significant to put forward the radiotracer as a novel imaging tool for prostate cancer. However, the authors need to critically review and proofread the manuscript for grammatical and typo errors before it can be accepted for publication. In addition to this, please find below some sentences that require editions.

Line 23: Grammatical error with a word missing between “was” and “in”.

Line 80: Edit “complete” to “completed”

Line 133: Edit “Treated” to “treating”

Line 136: End the sentence with “,respectively”.

 Furthermore, it could be of more interest to the readers if the authors could briefly discuss the currently available radiotracers in prostate and other cancers as well as any such imaging tools currently in the clinical pipeline.

Author Response

We thank the Reviewer 2 for his(her) overall positive assessment of our work and for constructive criticism. Please find our point-by-point responses to the reviewer’ comments below.

Reviewer 2 The authors address a critical problem pertaining to prostate cancer imaging and have therefore designed and evaluated a novel PSMA targeted radiotracer. The authors have a good grasp on the methods performed and the manuscript is very well organized, has ideal studies performed and the results are significant to put forward the radiotracer as a novel imaging tool for prostate cancer. However, the authors need to critically review and proofread the manuscript for grammatical and typo errors before it can be accepted for publication.

We reviewed the manuscript for grammatical and typo errors. It was revised by MDPI proofreading services.

In addition to this, please find below some sentences that require editions.

Line 23: Grammatical error with a word missing between “was” and “in”. Done

Line 80: Edit “complete” to “completed” Done

Line 133: Edit “Treated” to “treating” Done

Line 136: End the sentence with “,respectively”. Done

Furthermore, it could be of more interest to the readers if the authors could briefly discuss the currently available radiotracers in prostate and other cancers as well as any such imaging tools currently in the clinical pipeline.

As requested, we added further discussion on the currently clinically available PET radiotracers in prostate cancer (page 11 of 19).